# Exclusion of Mucorales Co-Infection in a Patient with *Aspergillus flavus* Sinusitis by Fluorescence In Situ Hybridization (FISH)

**DOI:** 10.3390/jof8030306

**Published:** 2022-03-16

**Authors:** Johanna Kessel, Michael Hogardt, Lukas Aspacher, Thomas A. Wichelhaus, Jasmin Gerkrath, Emely Rosenow, Jan Springer, Volker Rickerts

**Affiliations:** 1Department of Internal Medicine, Infectious Diseases, University Hospital Frankfurt, Goethe University, 60590 Frankfurt am Main, Germany; 2Institute for Medical Microbiology and Infection Control, University Hospital Frankfurt, Goethe University, 60590 Frankfurt am Main, Germany; michael.hogardt@kgu.de (M.H.); thomasa.wichelhaus@kgu.de (T.A.W.); 3Department of Internal Medicine, Hematology/Oncology, University Hospital Frankfurt, Goethe University, 60590 Frankfurt am Main, Germany; lukas.aspacher@kgu.de; 4Robert Koch Institute Berlin, FG16, Seestrasse 10, 13353 Berlin, Germany; gerkrathj@rki.de (J.G.); rosenowe@rki.de (E.R.); rickertsv@rki.de (V.R.); 5Medizinische Klinik und Poliklinik II, Universitätsklinikum Würzburg, 97080 Wuerzburg, Germany; springer_j@ukw.de

**Keywords:** invasive fungal infection, fluorescence in situ hybridization (FISH), fungal sinusitis, *Aspergillus*, mixed infection

## Abstract

Invasive fungal infections are associated with increased mortality in hematological patients. Despite considerable advances in antifungal therapy, the evaluation of suspected treatment failure is a common clinical challenge requiring extensive diagnostic testing to rule out potential causes, such as mixed infections. We present a 64-year-old patient with secondary AML, diabetes mellitus, febrile neutropenia, and sinusitis. While cultures from nasal tissue grew *Aspergillus flavus*, a microscopic examination of the tissue was suggestive of concomitant mucormycosis. However, fluorescence in situ hybridization (FISH) using specific probes targeting *Aspergillus* and Mucorales species ruled out mixed infection. This was confirmed by specific qPCR assays amplifying the DNA of *Aspergillus*, but not of Mucorales. These results provided a rational basis for step-down targeted therapy, i.e., the patient received posaconazole after seven days of calculated dual therapy with liposomal amphotericin B and posaconazole. Despite clinical response to the antifungal therapy, he died due to the progression of the underlying disease within two weeks after diagnosis of fungal infection. Molecular diagnostics applied to tissue blocks may reveal useful information on the etiology of invasive fungal infections, including challenging situations, such as with mixed infections. A thorough understanding of fungal etiology facilitates targeted therapy that may improve therapeutic success while limiting side effects.

## 1. Introduction

Patients with hematological malignancies and prolonged neutropenia are at increased risk of invasive fungal infections (IFI) [1,2]. Incidences of IFI are 5–15% among patients under hematological stem cell transplantation (HSCT) [3], even in the era of mold-active antifungal prophylaxis [4,5]. Therefore, IFI are of differential diagnostic concern in patients presenting with febrile neutropenia not responding to broad-spectrum antibiotic therapy [6].

The diagnosis of IFI [7] is challenging since the availability of diagnostic methods is often limited and the performance of tests when attempting to diagnose the whole spectrum of IFI, including mixed fungal infections, is suboptimal [8,9].

The most common site of infection is the lung, followed by nasal sinuses, depending on the causative fungal pathogen [2,10]. Since broad-spectrum azole prophylaxis is widespread in adult hematological patients, breakthrough infections are of special therapeutic concern. The outcome depends on the timely initiation of an adequate antifungal treatment and the control of the underlying hematological malignancy [11,12,13]. Despite considerable advances in antifungal therapy, the evaluation of potential treatment failure is a common clinical situation, with extensive diagnostic testing used to rule out potential reasons for treatment failure, including mixed fungal infections [14]. Such infections are diagnosed in up to 8% of patients with fungal sinusitis [15]. The confirmation of fungal etiology might be achieved through invasive tissue sampling, including sinus or lung biopsies. However, as cultures may fail to grow fungi and the microscopic identification of fungal elements in tissue may be insufficient to identify the infecting species, broad-range PCR and the sequencing of amplicons is recommended to identify fungal pathogens from tissue blocks [16]. However, this approach may fail in cases of fungal coinfections, as mixed amplicons cannot be identified via Sanger sequencing. Therefore, additional molecular diagnostic approaches are needed.

We report on a patient with culture-confirmed sinusitis due to *A. flavus*. When the failure of empiric antifungal therapy was suspected and histopathology suggested concomitant mucormycosis, fluorescence in situ hybridization was applied to the sinus tissue and allowed us to confirm aspergillosis and to exclude mucormycosis through the hybridization of an *Aspergillus*-specific probe, but not a Mucorales-specific probe, to all fungal elements present in the sample.

## 2. Case Report

A 64-year-old male patient was diagnosed with secondary acute myelogenous leukemia (AML) in November 2017 after exhibiting myelodysplastic syndrome (MDS). Comorbidities included coronary heart disease with consecutive cardiac insufficiency, renal insufficiency, and diabetes mellitus. Therefore, allogeneic stem cell transplantation and intensive chemotherapy were avoided as a therapeutic option. Instead, antineoplastic therapy was initiated with azacitidine and was switched to decitabine after three cycles because of allergic reactions. After four cycles of decitabine, he made hematological progress and was switched to venetoclax plus cytarabine. The patient developed persisting neutropenia (<1500 neutrophils/µL) with intermittent phases of severe neutropenia (<500 neutrophils/µL).

The patient presented nine months after diagnosis of AML with febrile neutropenia and a two-week history of photophobia, watery eyes, greyish rhinorrhea, and painful sinuses. His heart rate was 80 bpm, blood pressure 130/80 mmHg, and temperature 38.3 °C, and he had no respiratory limitations and no further neurological symptoms. Three sets of blood cultures, urine cultures, a PCR for influenza A/B and respiratory syncytial virus from a throat swab, and cultures from a nasal swab were negative. On the first day in hospital (day 1), empiric antibacterial therapy was started with piperacillin/tazobactam without a resolution of infection. A CT scan of the nasal sinuses on day seven showed mucosal swelling and a bone affection of the ethmoidal cells without signs of CNS affection. A thoracic CT scan revealed a nodular lesion of 1.3 × 1.1 cm in the right upper lobe, suggestive of a disseminated invasive infection (Figure 1). Empiric dual antifungal therapy was started with liposomal amphotericin B 3 mg/kg OD instead of 5 mg/kg OD, since renal function was impaired, and posaconazole, 300 mg OD (after loading with 300 mg BID), to minimize the risk of treatment failure. Defervescence occurred within 48 h, followed by a decrease in the amount of C-reactive protein (CRP). After an increase in serum creatinine in pre-existing renal insufficiency, renal function remained stable with a calculated GFR between 25 and 38 mL/min/1.73 m^2^.

In a sputum sample from day 7, microscopy with an optical brightener revealed mold hyphae, but only *Candida* spp. could be cultured. From nasal swabs, also only *Candida* spp. grew. On day 9, intranasal biopsies were performed to identify causative microbes to guide subsequent antifungal management. Histopathology demonstrated diverse fungal elements suggestive of a mold sinusitis with broad pauci-septate hyphae suggestive of mucormycosis and thin, septate hyphae suggestive of hyalohyphomycosis, such as aspergillosis (Figure 2).

Microbiological cultures grew *Aspergillus flavus*, which was tested for its antifungal susceptibility via microdilution (Sensititre Yeast One; Thermo Fisher, Wesel, Germany). According to the currently available EUCAST breakpoints for *A. flavus*, the isolate was susceptible to itraconazole and isavuconazole [17]. As no further clinical breakpoints for *A. flavus* are available, those defined for *A. fumigatus* may be used. A corresponding interpretation of measured MIC values classified voriconazole and posaconazole as susceptible and amphotericin B and caspofungin as resistant [17].

The *Aspergillus* antigen (GM EIA) in serum on day 6 showed an elevated value (OD 0.52).

The pathology block of the sinus tissue was used for fungal identification, i.e., to exclude the presence of a mixed-species infection. Molecular methods, including specific qPCR assays and fluorescence in situ hybridization, were performed and confirmed aspergillosis but ruled out concomitant mucormycosis through the use of specific probes. Antifungal therapy was reduced to posaconazole tablets (300 mg OD). The patient was readmitted six days later with a massive increase in lactate dehydrogenase, a deterioration of his general state, hypotension, and renal failure, most likely due to the progression of the hematological disease. There were no clinical signs of progress of the sinusitis or the pneumonia. He received the best supportive care. The patient died of multiorgan failure two days later.

## 3. Molecular Diagnostics Applied to the Formalin-Fixed, Paraffin-Embedded (FFPE) Sinus Tissue

The formalin-fixed, paraffin-embedded paranasal sinus tissue was cut in 5 µm sections with a Leica SM2010 R microtome (Leica Biosystems GmbH, Wetzlar, Germany), and placed on HistoBond^®^ + Adhesive Microscope Slides (Paul Marienfeld GmbH & Co. KG, Lauda-Königshofen, Germany). For the visualization of fungal elements, the GROCOTT’s Staining Kit for Fungi (Morphisto GmbH, Offenbach, Germany) was used according to the manufacturer’s instructions. FISH was performed as described previously after paraffin was removed from the unstained slides by dipping them in 99% octane for 15 s at room temperature and then air-drying them [18]. Subsequently, 100 µL hybridization buffer containing 100 ng of probe (Table 1) and 50 ng of DAPI were applied to each slide. As agents of mucormycosis are phylogenetically diverse, a combination of probes, one termed “mucor”, targeting most agents of mucormycosis, except for Lichtheimia, and a probe termed “Lichtheimia” that targets Lichtheimia sp. were used.

The buffer consisted of 5x SET (0.75 M NaCl, 5 mM EDTA, 0.1 M Tris (pH 7.8)), 10% dextran, 0.2% BSA, 0.1 mg/mL poly-adenosine, 20 µg/mL double-stranded salmon testes DNA, and 0.02% SDS. After the placement of coverslips, the slides were incubated over night at 50 °C in a humid chamber in the dark. Hybridization was stopped and coverslips were removed by placing the slides in 4 °C 5x SET, then washing them three times in 0.2x SET for 10 min for each washing step. VECTASHIELD^®^ Antifade Mounting Medium (H-1000-10) (Vector Laboratories, Inc., Burlingame, CA, USA) and a coverslip were applied to maintain fluorescence over longer periods. The coverslips were sealed with nail polish to prevent the leakage of the mounting medium. The correct performance of the hybridization procedure was controlled by target and non-target fungi for each probe. Fungi were grown at 37 °C for 6–8 h in RPMI or over-night under constant agitation in Sabouraud glucose broth. Liquid cultures were centrifuged, the supernatant was discarded, and the pellets were washed with PBS. Fungi were fixed in 4% paraformaldehyde (PFA) for 1 h at room temperature, followed by three washing steps with PBS and storage in 50% ethanol at 4 °C until use. Controls were spotted onto 8-well microscope slides (Thermo Fisher Scientific Inc., Waltham, MA, USA), with 10 µL per fungus. Hybridization was carried out as described above. Slides containing the sinus tissue or cultivated fungi were examined through epifluorescence microscopy with a Zeiss Axio Imager.Z1 with filter sets FS38 HE, FS43 HE, FS49, and FS50, and pictures were taken with the Zeiss AxioCam MR R3 (all from Carl Zeiss AG, Oberkochen, BW, Germany). For tissue samples, the Zeiss ApoTome.2 (Carl Zeiss AG, Germany) was used. Fluorescence signal intensity was visually checked with set exposure times, 90 ms for Cy3, Alexa Fluor 488 and DAPI, and 500 ms for Cy5. The fluorescence of tissue samples was compared to the fluorescence signal achieved with positive and negative controls. A fluorescence signal was considered to be positive if it was stronger than the negative control of the same fluorophore. Fluorescence signals with nonsense probes or with wavelengths not matching the fluorophores were considered to be non-specific or as autofluorescence.

Controls documented the correct performance of FISH, i.e., specific hybridization of probes on *Aspergillus* or Mucorales fungi only without hybridization of non-target fungi by these probes (Figure 3A). FISH results on the tissue showed hybridization of all fungal elements with the *Aspergillus* and unspecific probe but not with the Mucorales probe, suggesting invasive aspergillosis without concomitant mucormycosis, contrary to the Grocott’s staining showing narrow and broad mold hyphae in this diabetic neutropenic patient.

In order to control these results, fungal DNA was extracted from three cuts (5 µm each) of the tissue block, as described previously, and amplified using broad-range PCR targeting the 28S gene, which was performed as described previously [19]. This PCR amplified the DNA of different fungi preventing amplicon identification by sanger sequencing. Therefore, specific qPCR assays targeting the ITS region of *Aspergillus* and the 18S gene of Mucorales were applied [20]. These assays amplified the DNA of *A. flavus*, as identified by amplicon sequencing. The absence of amplifiable Mucorales DNA confirms the result of FISH, demonstrating that all FISH-positive hyphae only hybridize with the *Aspergillus* but not with the Mucorales probe.

## 4. Discussion

Invasive fungal infections are relevant infectious complications in hematological patients and are often associated with worse outcomes [21]. Defining the etiology of invasive mold infections is difficult but of paramount importance for optimal therapeutic management, including antifungal and surgical therapy [22]. The evaluation of potential treatment failure is a common clinical situation requiring extensive diagnostic testing to rule out potential reasons, including mixed infections [14]. Especially in suspected breakthrough infections and if non-Aspergillus mold infections are suspected, a quick diagnostic approach is needed [23]. Coinfections due to different molds are considered to be a relevant reason for suboptimal response to antifungal therapy [14].

Although the incidence of mixed infections is difficult to assess due to diagnostic challenges, mixed infections by *Aspergillus* and Mucorales have been diagnosed through conventional diagnostic approaches in 8% of patients with fungal sinusitis [15,24]. Of note, with the application of specific PCR tests, the DNA of both *Aspergillus* and Mucorales was detected in 2 of 27 (7%) samples [15].

In our patient, acute sinusitis due to *A. flavus* was evident from the cultivation of this fungus from sinus biopsy and positive serum GM. The presence of diverse fungal elements in the sinus biopsy, together with a mixed amplicon in a broad-range fungal PCR, required the exclusion of a coinfection with Mucorales in this diabetic neutropenic patient before the de-escalation of antifungal therapy could be initiated. As cultures frequently fail to grow Mucorales, we applied a molecular strategy to the FFPE sinus tissue using previously described specific FISH probes targeting *Aspergillus* and the agents of mucormycosis [25,26]. These probes were applied together with an unspecific probe targeting eukaryotic cells using different fluorophores. Using this approach, we could demonstrate that all fungal elements demonstrating a signal with the unspecific probe also showed a signal with the *Aspergillus*, but not the Mucorales probe, suggesting the absence of Mucorales hyphae in the sample. This was confirmed by the detection of DNA of *A. flavus* but not of agents of Mucorales in specific qPCR assays, as described previously [20]. This documents a substantial morphologic variability of hyphal elements in tissue sections in mold infections that may cause diagnostic uncertainty with the need for further diagnostic tests. In this situation, broad-range PCR with amplicon sequencing is the approach that experts recommend [16]. However, the presence of fungal DNA from more than one species, as demonstrated in our case, may preclude the identification of a fungal pathogen. While specific PCRs are more sensitive than broad-range PCRs, they only detect targeted organisms, and therefore do exclude additional fungal pathogens [20]. As broad-range assays may amplify the DNA of multiple agents, next-generation sequencing (NGS) might be required to identify amplified fungal DNA [27]. However, in non-sterile sites such as the paranasal sinuses, the DNA of multiple fungal species, invasive or colonizing, might be detected, necessitating an interpretation of sequencing results to deduce fungal etiology. In contrast, FISH, using differentially labeled specific and unspecific probes, may identify common pathogens for which specific probes are available and also additional pathogens by labeling with an unspecific probe only. A comparable approach might be possible using immunohistochemistry with antibodies targeting specific fungi [28].

## 5. Conclusions

We used a novel FISH approach including differentially labelled specific and unspecific probes to exclude *Aspergillus* and Mucorales coinfection, as suggested by suggestive fungal elements in a sinus biopsy with mixed fungal DNA amplified by a broad-range PCR assay in a diabetic neutropenic patient with refractory AML in order to de-escalate antifungal therapy. A specific PCR for *Aspergillus spp*. and FISH were conclusive in this case, suggesting that the FISH strategy might provide valid, additional insights when used in combination with other tools in FISH positive samples.

## Figures and Tables

**Figure 1 jof-08-00306-f001:**
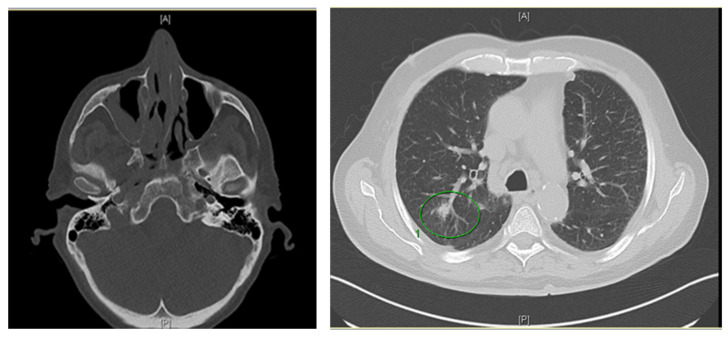
Radiological findings in a patient with acute myelogenous leukemia demonstrating mucosal swelling of the right sinus maxillaris and a nodular lung lesion (green circle) suggesting sinusitis and pneumonia.

**Figure 2 jof-08-00306-f002:**
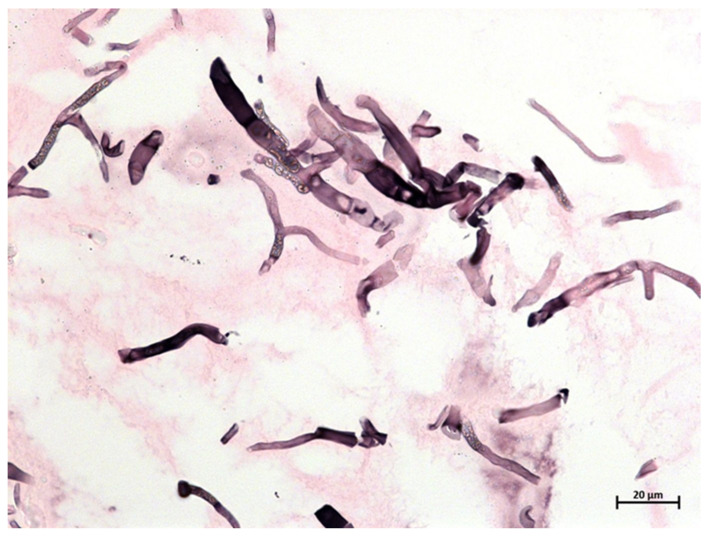
Grocott’s methenamine silver staining of sinus mucosal biopsy demonstrating diverse fungal elements, including broad, pauci-septate hyphae (5–10 µm) in line with mucormycosis and thinner hyphae, mostly with right-angle branching suggestive of hyalohyphomycosis. Of note, the sequencing of a positive broad-range fungal PCR demonstrated a mixed amplicon suggestive of the presence of different fungi and, therefore, preventing the identification of a single causative fungal pathogen.

**Figure 3 jof-08-00306-f003:**
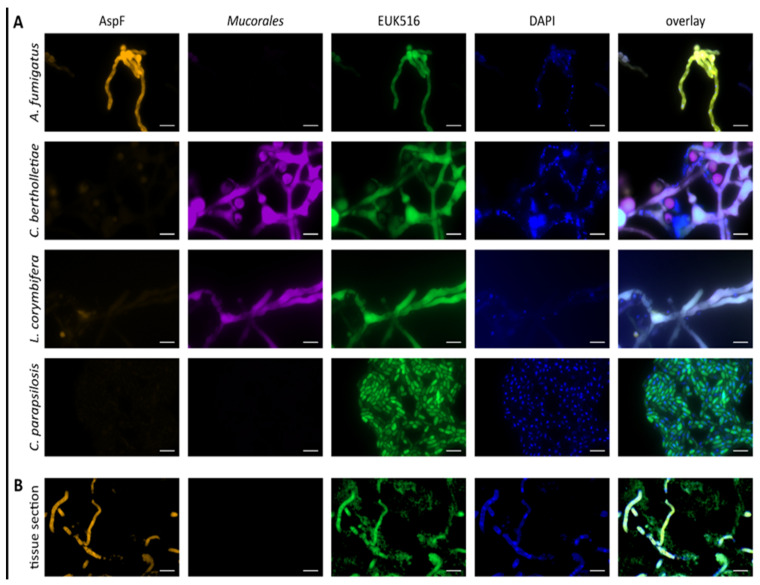
Fungal rRNA FISH of paranasal tissue biopsy and controls. (**A**) Fungi fixed in formalin reveal specific hybridization of probes targeting *Aspergillus* (AspF, labeled with Cy3, colored orange), the agents of mucormycosis (Mucorales, an equimolar mixture of the probes mucor and Lichtheima labeled with Cy5, colored magenta), and control probes targeting eukaryotic cells (labeled with Alexa fluor488, colored green). Results of nonsense probes are not shown. Counterstaining with DAPI was used to visualize double-stranded DNA in nuclei. (**B**) Hyphal elements of different width hybridizing exclusively with the probe targeting Aspergillus and the unspecific control probe but not the Mucorales probe, suggesting aspergillosis without evidence for concomitant mucormycosis. Note scant fluorescence signal of hyphae in tissue sample in the DAPI channel as compared to controls. Bar represents 10 µm.

**Table 1 jof-08-00306-t001:** List of used FISH probes: pos: positive control, neg: negative control, AspF: *Aspergillus* spp., EUK516: unspecific eukaryotic probe, nonEUB: nonsense probe, Muc: probe targeting Mucorales except for Lichtheimia, Lichtheimia: probe targeting Lichtheimia. These two probes were used in combination.

Probe Name	Target	Use	Sequence (5′->3′)	Dye	Tm (°C)
mucor	18S	Muc	CACGTACTTTTTCACTCTC	5′-Cy5	52.4
Lichtheimia	18S	Muc	GCTTTAAACACTCTGATTTG	5′-Cy5	51.5
AspF	28S	Asp	TGACGGCCCGTTCCAG	5′-Cy3	56.9
EUK516	18S	pos	ACCAGACTTGCCCTCC	5′-AF488	54,3
nonEUB	-	neg	ACTCCTACGGGAGGCAGC	5′-AF488	60.5
				5′-Cy3	60.5
				5′-Cy5	60.5

## Data Availability

Not applicable.

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
