# Peer review of "Exclusion of Mucorales Co-Infection in a Patient with Aspergillus flavus Sinusitis by Fluorescence In Situ Hybridization (FISH)"

_jof, 2022, doi:10.3390/jof8030306_

Round 1
Reviewer 1 Report
Comment to the authors
The article by Kessel et al. describes a case report of invasive fungal infection with sinusitis caused by A. flavus in an AML patient associated with febrile neutropenia. The authors also emphasize the usefulness of the FISH method applying to FFPE specimens to find out or exclude co-infection of invasive mold infection. It is an interesting report and has merit for readers, but some revision is essential.
Major comments:
Line 114-117
The histopathology of lung biopsy seems to indicate a diagnosis of invasive mold infection. Is it possible to present a histopathology picture?
It is challenging to diagnose when the amount of the fungal element is only few in the biopsy specimen. However, is there any form suggesting Mucor in the lung biopsy? Is there any difference between from sinusitis specimen? This point is important when the authors think about the co-infection of invasive fungal infection. Please describe clearly.
Line 122 and Line 233
The author concludes that the patient died from the original disease, not an invasive fungal infection. It would be better to mention more detail of the cause of death (Respiratory failure? Heart failure?) to say the success of de-escalation of antifungal therapy. Did the authors follow the pneumonia passage?
Minor comments:
Line19 and Line 21
A literature citation is provided in the abstract. As for the Abstract, it should not have any literature citations and should be revised.
Line 105
Did the authors try any drug sensitivity testing to A.flavus, which grown in the cultures? It would be more valuable as a case report if the authors could show some of the results of antifungals. The data also can support the reactivity of antifungals used.
Line 137
Is there any comment on the result of Lictheimia FISH probe? I could not find it in the text.
Line 189
A typo miss in the document. Remove period.
Author Response
Dear reviewer,
we thank you for your comments and suggestions which were very helpful to improve our manuscript. Please find our answers point-by-point below.
Yours sincerely,
J. Kessel
Major comments:
Line 114-117
The histopathology of lung biopsy seems to indicate a diagnosis of invasive mold infection. Is it possible to present a histopathology picture?
To our very discomfort, after careful consideration, we have to state a mistake in the manuscript at this point. One of the tissue biopsies that was sent to the microbiology lab was misinterpreted as lung biopsy, but actually it was nasal sinus biopsy. Since Aspergillosis could be confirmed from nasal biopsy and the patient was in reduced general condition, an additional lung biopsy has not been performed at this timepoint. Unfortunately, we did not recognize this error before submission. We really apologize for this. The main statement of the manuscript, however, should not be affected by the lack of a lung biopsy.
It is challenging to diagnose when the amount of the fungal element is only few in the biopsy specimen. However, is there any form suggesting Mucor in the lung biopsy? Is there any difference between from sinusitis specimen? This point is important when the authors think about the co-infection of invasive fungal infection. Please describe clearly.
Please see explanation above. We agree with the reviewer that it is challenging to diagnose an invasive fungal infection when the amount of fungal elements is limited, as in the reported case. In the sinus biopsy presented, mold hyphae clearly differ in width. Other “typical” morphological signs for aspergillosis and mucormycosis are lacking, which is often the case when tissue of limited size is available. Therefore, FISH was applied, which only showed evidence for aspergillosis. This suggests that FISH might be useful to exclude mixed fungal infections.
Line 122 and Line 233
The author concludes that the patient died from the original disease, not an invasive fungal infection. It would be better to mention more detail of the cause of death (Respiratory failure? Heart failure?) to say the success of de-escalation of antifungal therapy. Did the authors follow the pneumonia passage?
The patient presented with general weekness and generalized pain. His Lactate dehydrogenase- values were sixfold higher than four days ago and progression of his leucemia was highly suspected.Progressive symptoms of sinusitis or pneumonia were not identified. He was hypotensive and renal function was very impaired. Since there were no therapeutic options to control leucemia, he received best supportive care with pain control and supplementary oxygen later on. We refused to do any further CT scans. The patient died of multiorgan failure in progression of his leucemia. We added these findings in the manuscript in lines 128-133.
Minor comments:
Line19 and Line 21
A literature citation is provided in the abstract. As for the Abstract, it should not have any literature citations and should be revised.
Citations are revised. Thank you for the hint. As it would be very confusing to show all literature citations in the new order in track change mode, we changed the citations according to the new sequence in normal mode and added all changes in der man text in track change mode thereafter.
Line 105
Did the authors try any drug sensitivity testing to A.flavus, which grown in the cultures? It would be more valuable as a case report if the authors could show some of the results of antifungals. The data also can support the reactivity of antifungals used.
Yes, we did. Results were added in the manuscript Line 106-112.
Line 137
Is there any comment on the result of Lictheimia FISH probe? I could not find it in the text.
We thank the reviewer for this comment. As the agents of mucormycosis are phylogenetically diverse, we used a probe-combination to target the most prevalent agents of mucormycosis. Both probes are listed in table 1 (“mucor” and Lichtheimia”). This information is now given in lines 144-146, 149 and 194-197.
Therefore, we did not see a hybridization signal with both probes (mucorales and Lichtheimia) used together.
Line 189
A typo miss in the document. Remove period.
Done, thank you.
Reviewer 2 Report
In this paper the authors describe a case of invasive fungal infection with the aim of promoting fluorescent in situ hybridization (FISH) to exclude a mixed infection.
The paper is clear, well-written and the case very well-documented. Arguments are robust.
The case concerns an immunocompromised patient (acute leukemia) who develops febrile sinusitis with a pulmonary nodule on the CT-scan. Intranasal biopsies reveal fungal hyphae some of which being tortuous and pauci-septate, making the diagnosis of mixed aspergillosis-mucormycosis possible. Culture further grows positive for Aspergillus fumigatus.
FISH was performed on these specimens with specific probes targeting Aspergillus and mucorales. Adequate controls were also used.
Furthermore, specific PCR for Aspergillus and mucorales were applied to distinguish between Aspergillus infection alone or mixed Aspergillus-Mucorales infection.
Both results of FISH and PCR advocate for Aspergillosis alone.
While the conclusion drawn by the authors is (very) highly probable, I think it is quite difficult to conclude that in every other case the conclusion would be the same, notably in the case of negative culture.
First the sensitivity and specificity of the specific qPCRs are, as it is the rule, not 100% (Springer et al 2016 & 2019)
Second it cannot be excluded that looking at another part of the biopsy specimen stained with FISH, one can find some hyphae that would be detected with the Mucorales probe. This may be one of the arguments of the EORTC-MSGERC that concludes for hybridization that « Although promising, more work needs to be done before it can be recommended ».
Overall, it always seems difficult to firmly excluded an infection based on negative results. A more conditionnal form in the conclusion would be appropriate
Author Response
Dear reviewer,
many thanks for the your comments and suggestions which hepled to improve our manuscript. We answered to your comment below.
Yours sincerely,
J. Kessel
Reviewer 2:
In this paper the authors describe a case of invasive fungal infection with the aim of promoting fluorescent in situ hybridization (FISH) to exclude a mixed infection.
The paper is clear, well-written and the case very well-documented. Arguments are robust.
The case concerns an immunocompromised patient (acute leukemia) who develops febrile sinusitis with a pulmonary nodule on the CT-scan. Intranasal biopsies reveal fungal hyphae some of which being tortuous and pauci-septate, making the diagnosis of mixed aspergillosis-mucormycosis possible. Culture further grows positive for Aspergillus fumigatus.
FISH was performed on these specimens with specific probes targeting Aspergillus and mucorales. Adequate controls were also used.
Furthermore, specific PCR for Aspergillus and mucorales were applied to distinguish between Aspergillus infection alone or mixed Aspergillus-Mucorales infection.
Both results of FISH and PCR advocate for Aspergillosis alone.
While the conclusion drawn by the authors is (very) highly probable, I think it is quite difficult to conclude that in every other case the conclusion would be the same, notably in the case of negative culture.
First the sensitivity and specificity of the specific qPCRs are, as it is the rule, not 100% (Springer et al 2016 & 2019)
Second it cannot be excluded that looking at another part of the biopsy specimen stained with FISH, one can find some hyphae that would be detected with the Mucorales probe. This may be one of the arguments of the EORTC-MSGERC that concludes for hybridization that « Although promising, more work needs to be done before it can be recommended ».
Overall, it always seems difficult to firmly excluded an infection based on negative results. A more conditionnal form in the conclusion would be appropriate
We agree with the reviewer on all the limitations of molecular tests. However, mixed infections do occur and diagnosis can be puzzling. Specific qPCRs are probably the most sensitive approach to exclude the presence of amplifiable DNA of fungi targeted by a PCR assay. Broadrange PCRs, endorsed by the EORTC-MSG guideline for fungal identification from FFPE tissues may fail to identify fungal DNA due to mixed amplification. Therefore, remaining molecular options may include FISH with a combination of specific and unspecific probes as we are suggesting here and ngs.
We do not advocate the use of FISH as a standard approach to identify fungi in FFPE tissues but this approach may be helpful in some cases. In line with the reviewers suggestions, we toned the conclusion down (line 260-262).